



# Estimation of cloud optical thickness, single scattering albedo and effective droplet radius using a shortwave radiative closure study in Payerne

Christine Aebi[1,2,*], Julian Gröbner[1], Stelios Kazadzis[1], Laurent Vuilleumier[3], Antonis Gkikas[4], and Niklaus Kämpfer[2]

[1]Physikalisch-Meteorologisches Observatorium Davos, World Radiation Center, Davos, Switzerland.
[2]Oeschger Center for Climate Change Research and Institute of Applied Physics, University of Bern, Bern, Switzerland.
[3]Federal Office of Meteorology and Climatology MeteoSwiss, Payerne, Switzerland.
[4]Department of Physics, University of Ioannina, Ioannina, Greece.
[*]now at: Royal Meteorological Institute of Belgium, Brussels, Belgium.

**Correspondence:** Aebi Christine (christine.aebi@pmodwrc.ch)

**Abstract.** We have used a method based on ground-based solar radiation measurements and radiative transfer models (RTM) in order to estimate the following cloud optical properties: cloud optical thickness (COT), cloud single scattering albedo (SSAc) and effective droplet radius ($r_{eff}$). The method is based on the minimisation of the difference between modelled and measured downward shortwave radiation (DSR). The optical properties are estimated for more than 3,000 stratus-altostratus (St-As) and 206 cirrus-cirrostratus (Ci-Cs) measurements during 2013-2017, at the Baseline Surface Radiation Network (BSRN) station in Payerne, Switzerland. The RTM libRadtran is used to simulate the total DSR, as well as its direct and diffuse components. The model inputs of additional atmospheric parameters are either ground- or satellite-based measurements. The cloud cases are identified by the use of an all-sky cloud camera. For the low- to mid-level cloud class St-As, 95 % of the estimated COT values from DSR measurements ($COT_{DSR}$) are between 11.9 and 91.5 with a geometric mean and standard deviation of 33.81 and 1.67, respectively. The comparison of these $COT_{DSR}$ values with $COT_{Barnard}$ values retrieved from an independent empirical equation, results in a mean difference of -1.20 ± 2.73 and is thus within the method uncertainty. However, there is a larger mean difference of around 18 between $COT_{DSR}$ and COT values derived from MODIS level-2 (L2), Collection 6.1 (C6.1) data ($COT_{MODIS}$). The estimated $r_{eff}$ (from liquid water path (LWP) and $COT_{DSR}$) for St-As are between 2.1 and 20.4 $\mu$m. For the high-level cloud class Ci-Cs, $COT_{DSR}$ is derived considering the direct radiation and 95 % of the values are between 0.32 and 1.40. For Ci-Cs, 95 % of the SSAc values are estimated to be between 0.84 and 0.99 using diffuse radiation measurements. The COT values for Ci-Cs are also estimated from data from precision filter radiometers (PFR) at various wavelengths. The herein presented method could be applied and validated at other stations with direct and diffuse radiation measurements.



# 1 Introduction

Clouds are a major component of the climate system and have a significant influence on the Earth's radiation budget (Boucher et al., 2013). Cloud optical thickness (COT) is a key parameter of the cloud optical properties, which in turn are of interest for determination of the cloud radiative effect (Jensen et al., 1994; Chen et al., 2000; Baran, 2009; Hong and Liu, 2015). The radiative properties of clouds are determined by their macrophysical and microphysical properties. The accurate parametrisation of these optical properties in climate models is a challenge because the small-scale physical processes of clouds are difficult to explicitly represent in global climate models (e.g. Waliser et al., 2009; Baran, 2012; Taylor, 2012; Zelinka et al., 2013; Ceppi et al., 2017). Thus, the introduction of methodologies using long-term observation aimed at the improvement of COT retrieval are important for estimating the magnitude of the influence of the diverse and variable cloud situations on the climate system.

A common practice to determine COT values is with the use of satellite-based instruments and the so-called bi-spectral method (Nakajima and King, 1990; Platnick et al., 2017). Although this approach has shown good results on a global scale, there are also a number of potential uncertainty sources, namely in the spectral radiation calibration, horizontal and vertical inhomogeneities and inappropriate use of cloud microphysics (Zeng et al., 2012). In addition, satellite-based lidar systems such as the Cloud-Aerosol Lidar with Orthogonal Polarization (CALIOP) provide high-resolution vertical profiles of clouds (Winker et al., 2009), including products such as cloud extinction and backscatter profiles (Amiridis et al., 2015). Other studies describe a COT retrieval method from satellites using neural network based approaches (Kox et al., 2014; Minnis et al., 2016).

Cloud optical properties can also be estimated from airborne measurements (e.g. Finger et al., 2016; Krisna et al., 2018). Flying directly below or above clouds allows both accurate measurements and direct comparisons and validations of the COT values retrieved from satellite sensors. However, these campaigns are cost-intensive and thus the spatial and temporal resolution of data is poor.

A number of studies have presented methods for the retrieval of COT using data from ground-based instruments, for example, from lidars (Gouveia et al., 2017), broadband pyranometers (Leontyeva and Stamnes, 1994; Barnard and Long, 2004; Qiu, 2006), sunphotometers (Min and Harrison, 1996; Chiu et al., 2010) or UV radiometers (Serrano et al., 2014). With ground-based microwave instruments the liquid water path (LWP) is determined (Dupont et al., 2018), which can be used to calculate the cloud optical thickness, knowing or assuming $r_{eff}$ (Stephens, 1994). Ground-based and airborne retrieval methods can be combined in order to achieve more accurate results for COT retrieval (Schäfer et al., 2018). Chiu et al. (2010) compared COT values retrieved from a sunphotometer with Moderate Resolution Imaging Spectroradiometer (MODIS) level-2 (L2) data with reasonable results in the COT agreement in few cases. In order to compare COT retrievals on a global scale, networks with large global coverage and density, as well as easily accessible data are needed. Barnard and Long (2004) showed a first approach in this direction by using only broadband diffuse shortwave radiation, albedo, solar zenith angle (SZA) and a clear sky model in order to estimate COT. It has been proven that this empirical equation can be used for homogeneous low-level, but not for high-level clouds. The aim of our study is to use a method which is based on a radiative closure study and which allows the determination of COT independent of the cloud level.

In principle, radiative closure studies assess the difference between modelled and measured shortwave or longwave radiation.





Among other things they allow estimation of the importance of accurate input variables and can be used to evaluate the accuracy of the retrieval of cloud optical properties (Wang et al., 2011). Nowak et al. (2008b) pointed out that in most cloud
cases, radiative closure can only be achieved by having information about the cloud microphysical properties. This because e.g. stratus nebulosus can have large variations in cloud extent, cloud droplet concentrations, optical thickness and liquid water path (Dong et al., 2000).

With data from an airborne measurement campaign, Ackerman et al. (2003) achieved an agreement in the global shortwave radiation of within 8 % to 14 % for three single-layered stratus cases only by iteratively determining $r_{eff}$. McFarlane and Evans
(2004) presented a study where they included $r_{eff}$ and liquid water content from microwave and cloud radar measurements in the model resulting in a difference of 10 % between simulated and measured global DSR. However, this fairly good agreement was only achieved after averaging the data over a time period of 60 minutes. Nowak et al. (2008b) achieved an agreement between the modelled and the observed shortwave radiation within measurement uncertainty in one third of 32 selected and well-defined stratus nebulosus cases without adjusting any cloud properties. For the other cases, the cloud vertical extinction
had to be adjusted in order to obtain an agreement within instrumental uncertainty. Wang et al. (2011) found a mean difference to within 5 % $\pm$13 % for shortwave radiation for more than 600 well-defined thick low-level cloud cases at the BSRN site Cabauw. They calculated COT according to the formula from Stephens (1994) using $r_{eff}$ from MODIS data and LWP from a ground-based microwave instrument.

In the current study we estimate $COT_{DSR}$ for stratus-altostratus (St-As) and cirrus-cirrostratus (Ci-Cs) using broadband short-
wave radiation measurements and ancillary ground- and satellite-based data from the BSRN station in Payerne, Switzerland, by performing a radiative closure study. The $COT_{DSR}$ for St-As and Ci-Cs are estimated using the diffuse and the direct component of DSR, respectively. For Ci-Cs, we show an attempt to estimate the SSAc from the diffuse component of DSR. The $r_{eff}$ for St-As is estimated from $COT_{DSR}$ and measured LWP by using the equation from Stephens (1994). Additionally, we investigate the sensitivity of the model input parameters as well as the robustness of the $COT_{DSR}$ and SSAc retrievals. Results
of such a model validation, combined with the measurement uncertainty, can provide the limits of the minimum possible agreement among modelled and measured solar radiation quantities under cloud-free and cloudy conditions. The retrieved $COT_{DSR}$ values are compared with $COT_{Barnard}$ values retrieved by applying the empirical equation from Barnard and Long (2004), $COT_{MODIS}$ values derived from MODIS L2 C6.1 data for different spatial resolutions and $COT_{PFR}$ values determined with a ground-based sun-pointing instrument.

The observational data and the case selection are presented in Section 2. Section 3 describes the radiative transfer, the RTM used and its input parameters, as well as the methods for the retrieval of the $COT_{DSR}$, SSAc and $r_{eff}$ values. In Section 4 the expanded combined uncertainty of the $COT_{DSR}$ and SSAc retrievals is estimated. Section 5 shows the obtained $COT_{DSR}$, SSAc and $r_{eff}$ values. Section 6 compares the $COT_{DSR}$ values with COT values determined using other methods. Finally, Section 7 summarises the main findings and gives a brief outlook.



## 2 Data

### 2.1 Observational Data

The aerological station of Payerne (46.49°N, 6.56°E, 490 m asl) is located in the western midlands of Switzerland between two mountain ridges. This station is managed by the Federal Office of Meteorology and Climatology (MeteoSwiss) and belongs to the BSRN (Ohmura et al., 1998; König-Langlo et al., 2013; Driemel et al., 2018). For the current study, high-accuracy radiation measurements from Payerne between January 1, 2013 and December 31, 2017 are used (Vuilleumier et al., 2014). The broadband downward shortwave radiation (DSR; 0.3 - 3 $\mu$m) is measured with a Kipp and Zonen CMP22 pyranometer. This instrument is traceable to the World Standard Group (WSG) located at the Physikalisch-Meteorologisches Observatorium Davos/World Radiation Center (PMOD/WRC) in Davos, Switzerland and measures within an uncertainty of 1 $\mathrm{Wm}^{-2}$ and a relative uncertainty of 2 %, whichever is larger (Vuilleumier et al., 2014). The diffuse and direct radiation values are measured with a CMP22 and a CHP1 pyrheliometer, respectively. The upward shortwave radiation (USR) is measured with a CMP21. All radiation data are corrected for the thermal offset (Philipona, 2002), homogenised and are available in a temporal resolution of 1 minute. The cloud base height (CBH) data are available in Payerne in a 1 minute temporal resolution from a CHM15k ceilometer (Wiegner and Geiß, 2012). The cloud fraction and cloud type are determined from images of an all-sky cloud camera (Schreder VIS-J1006), sensitive in the visible range of the spectrum, in a 5 minutes temporal resolution (Wacker et al., 2015; Aebi et al., 2017).

The aerosol optical depth (AOD) values at 550 nm wavelength are daily mean level-3 (L3) (Collection 6) data from the MODIS satellites Terra and Aqua (Kaufman et al., 1997). The overpasses over Europe are around 10:30 UTC ±1 hour (Terra) and around 13:30 UTC ±1 hour (Aqua). The horizontal resolution of these data is a 1°×1° grid cell. In order to validate these low spatial resolution data they are compared with ground-based AOD measurements from a precision filter radiometer (PFR; Wehrli (2000); Kazadzis et al. (2018)). In Payerne, for the cloud-free cases in the analysed time period, the mean difference in AOD between the two data sets is 0.00 ±0.07, showing that no significant bias between the two data sets is present. In Payerne, considering the MODIS AOD values, in the aforementioned time period, 90 % of the data have AOD values between 0.02 and 0.25, with lower values in winter than in summer (Figure 1a).

The integrated water vapour (IWV) is retrieved from GPS signals operated by the Federal Office for Topography. These data are then archived in the Studies in Atmospheric Radiation Transfer and Water Vapour Effects (STARTWAVE) database hosted at the Institute of Applied Physics at the University of Bern (Morland et al., 2006a). The 5th and 95th percentile values of the measured IWV values in Payerne are 6.0 mm and 30.6 mm, respectively. The values show a seasonal variation with larger values in summer than in winter (Figure 1b).

The total column ozone content is retrieved from the Ozone Monitoring Instrument (OMI) on the Aura satellite (Levelt et al., 2006, 2018). For Payerne, there are one to two data points available per day. The spatial resolution of the OMI total column ozone is 100 km in radius with Payerne in the center. Ozone data from the OMI satellite show good agreement with the results retrieved from ground-based Dobson and Brewer instruments at other stations (e.g. Vanicek, 2006; Antón et al., 2009). The total column ozone in Payerne has a seasonal cycle with high early spring and low autumn values (Figure 1c). The 5th and 95th



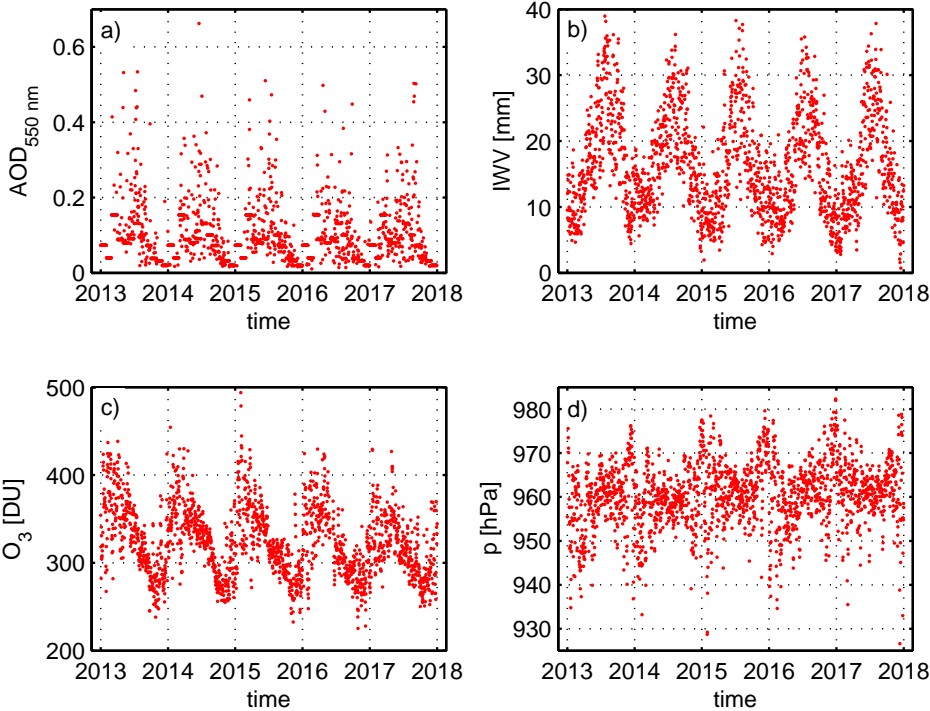

**Figure 1.** Time series of the following parameters: (a) aerosol optical depth (AOD), (b) integrated water vapour (IWV), (c) ozone ($O_3$) and (d) surface pressure ($p$) in Payerne.

percentiles are 266 and 397 DU, respectively.

The surface pressure is taken from a state-of-the-art measurement at the aerological station in Payerne. The mean surface pressure value is 960 hPa $\pm 7$ hPa (Figure 1d), with only small variations throughout the year.

Twice a day (at 12 and 00 UTC), the aerological station in Payerne launches a radiosonde, measuring among other parameters the profiles of pressure, temperature and relative humidity. The LWP is measured by a HATPRO (humidity and temperature profiler) instrument, also installed at MeteoSwiss in Payerne.

For the comparison of our $COT_{DSR}$ data, COT L2 C6.1 data from the MODIS instrument on the Aqua satellite are used (Nakajima and King, 1990; Platnick et al., 2017). These $COT_{MODIS}$ values are calculated using the measurements of the 1.64 $\mu$m channel of the MODIS instrument and are determined in the range 0 to 150. For our comparison we used $COT_{MODIS}$ data retrieved for grid points around Payerne with the dimensions of 3×3 km, 5×5 km and 7×7 km. From PFR data an optical thickness value under cirrus-cirrostratus conditions is determined and compared to the $COT_{DSR}$ values.





**Table 1.** Summary of the total number of measurements, number of days, cloud fraction considered and the occurrences per season for cloud-free, stratus-altostratus (St-As) and cirrus-cirrostratus (Ci-Cs) in Payerne.

| Cloud Class | Total # | # Days | Cloud Fraction | Occurrences [%] | | | |
|---|---|---|---|---|---|---|---|
| | | | | MAM | JJA | SON | DJF |
| Cloud-free | 13,240 | 379 | 0 - 1 % | 35.5 | 41.6 | 20.3 | 2.6 |
| St-As | 3,724 | 312 | 95 - 100 % | 60.1 | 6.2 | 21.2 | 12.5 |
| Ci-Cs | 206 | 48 | 95 - 100 % | 72.3 | 9.7 | 14.1 | 3.9 |

## 2.2 Case Selection

The herein presented analysis is shown for stratus-altostratus and cirrus-cirrostratus, which both have a distinct radiative behaviour. These two cloud classes were chosen due to their homogeneous cloud layer, along with the representation of a low- to mid-level water cloud class as well as a high-level ice cloud class. In order to validate the atmospheric model input variables, a shortwave radiative closure study is also performed for cloud-free conditions. The measurements were taken from the time period January 1, 2013 to December 31, 2017. The cloud coverage and the cloud type information are selected using images from the all-sky cloud camera in Payerne. From the RGB-information of the image, an automatic algorithm calculates a ratio per pixel which is subsequently compared to a reference threshold value. On the basis of this comparison it is decided whether a pixel is classified as cloudy or cloud-free. Also, the cloud classes are automatically determined from the images of the visible all-sky camera using an algorithm considering 12 spectral, textural and radiative features of the images (Wacker et al., 2015). All analysed data points have SZA values of maximum 65°. This maximum limit is defined in order to avoid cloud misclassifications due to the darker camera images that correspond to higher solar angles. Another reason for this threshold is the possible overestimation of the ground albedo estimation for high SZAs (Manninen et al., 2004).

Table 1 gives a summary of the number of measurements in total and per day, the cloud fractions considered and the occurrences throughout the year per cloud class separately. With 13,240 measurements spread over 379 days, the cloud-free data set is the largest. The visual checking of a part of the cloud-free data set allows the conclusion to be drawn that the cloud-free situations are determined with an accuracy of more than 90 %. The same number was also reported in Wacker et al. (2015). The distribution of these measurements per month is slightly different from the one reported in Aebi et al. (2017).

For St-As, in addition to the cloud fraction, further selection criteria are that the direct shortwave radiation does not exceed $1 \, \mathrm{Wm}^{-2}$ (in order to avoid cases of 95 % cloudiness but a clear solar path to the instrument) and that the CBH is equal to or below 5 km. Also the images of this data set were visually checked and the thick and homogeneous appearance of the cloud layer is confirmed in the remaining data set of 3,724 measurements.

As reported in Aebi et al. (2018), there are some uncertainties with the automatic detection of thin high-level clouds. Therefore, for the final data set of 206 measurements, only situations with a measured CBH of at least 5 km are considered. Additionally, the remaining cases were checked visually to avoid misclassifications. This data set is much smaller than the one for St-As clouds due to the fact that the occurrence of overcast Ci-Cs is less frequent in Payerne.





## 3 Methods

In the present study, we use radiative closure in shortwave radiation as a tool to retrieve COT values. The optical thickness ($\tau$)
is defined as the extinction ($\beta_e$) of radiation along a path from surface ($z_{surf}$) to the top of atmosphere ($z_{TOA}$),

$$\tau(z_{surf}, z_{TOA}) = \int_{z_{surf}}^{z_{TOA}} \beta_e(z)dz \tag{1}$$

where $\tau$ is the sum of the optical thickness of the different atmospheric components at a certain wavelength $\lambda$,

$$\tau(\lambda) = \tau_{cloud}(\lambda) + \tau_{AOD}(\lambda) + \tau_{IWV}(\lambda) + \tau_{O_3}(\lambda) + \tau_{Rayleigh}(\lambda) + \tau_g(\lambda) \tag{2}$$

where $\tau_{cloud}(\lambda)$ is the optical thickness of clouds (in this paper abbreviated as $COT$), $\tau_{AOD}(\lambda)$ the optical thickness of
aerosols, $\tau_{IWV}(\lambda)$ the optical thickness of water vapour, $\tau_{O_3}(\lambda)$ the optical thickness of ozone, $\tau_{Rayleigh}(\lambda)$ the optical thickness due to Rayleigh scattering and $\tau_g(\lambda)$ the optical thickness of other absorbing gases depending on the wavelength.

Assuming spherical droplets in a water cloud, the optical properties COT, SSAc and the asymmetry factor (which is the first moment of the droplet phase function), can be calculated from Mie theory (Stephens, 1978). However, assuming a homogeneous and plane-parallel water cloud layer, the SSAc and the phase function of the cloud droplets play a minor role in the
determination of the transmission of the cloud layer, in contrast to COT (Rawlins and Foot, 1990). Under this consideration, the shortwave radiative effect of a water cloud can be either characterised by the COT alone or by a combination of the $r_{eff}$ and the $LWC$ (Leontyeva and Stamnes, 1994). For the shortwave radiation range, the extinction coefficient in clouds, and thus also COT, has a weak dependence on the wavelength (Slingo and Schrecker, 1982). When $r_{eff}$ is increasing, the transmitted flux would decrease because of the larger absorption. However, at the same time, the transmitted flux would also increase
because of more forward scattering. The cloud droplet size distribution plays only a minor role in determining the extinction coefficient (Rawlins and Foot, 1990).

Whereas for thick water clouds the transmitted flux only comprises diffuse radiation, the transmitted flux for thin ice clouds comprises direct and diffuse radiation. In this case the Beer-Lambert law could be used to calculate the direct component of the shortwave radiation:

$$I(\lambda) = I_0(\lambda)e^{-m\tau(\lambda)} \tag{3}$$

where $\lambda$ is the wavelength, $I(\lambda)$ is the direct transmitted radiation at the surface and $I_0(\lambda)$ is the irradiance at the top of the atmosphere, $m$ the air mass and $\tau$ the sum of the optical thickness as shown in Equation 2. To determine the cloud optical properties of ice clouds, the microphysical properties of particle shape, particle size distribution and ice water content are of interest.

The single scattering albedo (SSA) is defined as the ratio between the scattering and total extinction coefficients and is wavelength dependent. The SSA is the weighted sum of the different components in the atmosphere, namely the single scattering albedo of clouds (SSAc), of aerosols, of molecules, etc. The SSAc is mainly of importance for the simulation of ice clouds



and its values differ depending on the size and shape of the ice crystals (Key et al., 2002). A more complete explanation of the relationships between the optical properties of water and ice clouds is given in Kokhanovsky (2004).


## 3.1 Radiative Transfer Model

The radiative transfer model libRadtran (library of radiative transfer routines and programmes) version 2.0.2 (Mayer and Kylling, 2005; Emde et al., 2016) is used to simulate the global DSR as well as the direct and the diffuse shortwave radiation. Our calculations use the discrete ordinate radiative transfer solver (DISORT) (Stamnes et al., 1988), which solves the one dimensional

plane-parallel radiative transfer equation. The number of streams is 6. Increasing the number of streams to the libRadtran maximum of 16 streams would result in a negligible difference in radiation estimations of less than 0.2 % in our calculations. The modelling is performed with the representative wavelength approach (REPTRAN) (Gasteiger et al., 2014) in a coarse resolution (15 cm$^{-1}$ band width). The calculated spectral range for DSR is 250 - 3,000 nm.

### 3.1.1 Input Parameters

The following atmospheric input parameters are defined for the libRadtran simulations:

Aerosols: For cloud-free cases, the AOD is a daily mean value from the two MODIS instruments at 550 nm. For cloudy conditions, or in cases of missing AOD values during cloud-free conditions, the AOD is a monthly mean value from MODIS data over the whole time period analysed. The aerosol profile is assumed to be a standard profile for a rural area described in Shettle

(1989) and the aerosol single scattering albedo value is assumed to be 0.95.

IWV: For all cases, the IWV is a daily mean (or if missing, the interpolated mean) value from GPS measurements in Payerne.

Ozone: The total column ozone is the daily mean (or if missing, the interpolated mean) of measured values of the OMI satellite.

Atmospheric profiles and surface pressure: The surface pressure is a daily mean value from measurements in Payerne. A standard mid-latitude atmospheric profile for either winter or summer is used with standard profiles of pressure, temperature, air

density and concentrations of different atmospheric gases (Anderson, 1986). The profiles are scaled to the input values IWV, ozone, surface temperature and pressure. The use of measured profiles of pressure, temperature and relative humidity from radiosondes has a negligible effect on the cloud-free diffuse radiation (0.3 %) and therefore the analyses are performed with the standard profiles.

Albedo: The shortwave surface albedo is calculated from the ratio of the USR to the DSR with 1-minute resolution. The mean

shortwave surface albedo is 0.24 with a variability of 0.15 and 0.45 (covering 90 % of the data) in the period analysed. In the few cases of snow the albedo can reach values up to 0.8. Due to the homogeneous landscape around the aerological station Payerne, the albedo derived from point measurements may be extrapolated to a larger region around the station.

SZA: The SZA is retrieved with a solar position algorithm for every measurement. The analysed data set includes SZA values between 23° and 65°.

Water clouds: The low- to mid-level St-As are water clouds simulated with the parametrisation described in Hu and Stamnes





(1993). They are assumed to be a plane-parallel and homogeneous cloud layer. The extinction coefficient for shortwave radiation is approximated from a vertical profile of LWC, $r_{eff}$ and the water density (Slingo and Schrecker, 1982). Assuming a constant LWC of 0.28 $gm^{-3}$ (Hess et al., 1998), $r_{eff}$ of 10 $\mu$m (Stephens, 1994), a cloud vertical thickness of 2 km and knowing the CBH from the ceilometer measurements results in a large relative mean difference (modelled minus measured divided by measured) and standard deviation of the global DSR of - 53.54 % $\pm$ 20.92 %, clearly demonstrating that the default values do not provide adequate results.

Ice clouds: The high-level Ci-Cs clouds are assumed to be complete ice clouds and are modelled with the parametrisation by Key et al. (2002). The optical property COT of ice clouds is parametrised using a vertical profile of ice water content (IWC) and effective ice crystal radius. The IWC is assumed to be 0.03 $gm^{-3}$ (Korolev et al., 2007; Navas-Guzmán et al., 2014) and the effective ice crystal radius 20 $\mu$m (Stubenrauch et al., 2013). The CBH is taken from ceilometer measurements and the cloud vertical thickness is assumed to be 1.5 km, which is a typical value for these high-level clouds (IPCC, 2013). As ice habit the default solid column is included. Using these default values to estimate the global DSR results in a relative mean difference of - 23.46 % $\pm$ 8.16 %, also demonstrating that these default values do not produce reliable results.

COT: For the simulation of the cloud cases, in addition to the profiles of LWC and $r_{eff}$, a COT value can also be explicitly defined as input to the model. To iteratively derive the effective $COT_{DSR}$, we used COT as input.

## 3.2 COT, SSAc and $r_{eff}$ retrieval

The aim of our study is to determine COT, SSAc (for Ci-Cs) and $r_{eff}$ (for St-As). In order to retrieve COT and SSAc, we derive the total DSR as well as its components, direct and diffuse radiation, from libRadtran. The DSR is calculated for every single measurement point in the cloud data sets separately. Lookup tables (LUT) are generated, containing the simulated radiation values, which were modelled with different COT and SSAc values as input, respectively. More precisely, every measurement point in for example the St-As data set, is simulated with input to libRadtran of 25 different COT values between 1 and 160. Since in the case of St-As the direct component of the total DSR is smaller than 1 $\mathrm{Wm}^{-2}$, the St-As COT LUT contains 3,724 times 25 simulated diffuse radiation values. This LUT serves thereafter to estimate $COT_{DSR}$ for all the 3,724 St-As cloud cases separately by finding the simulated diffuse radiation value that equals the measured one.

The $COT_{DSR}$ for Ci-Cs is estimated with a similar method. The only difference is that this LUT contains simulated direct radiation values, which were estimated with 13 different COT input values between 0 and 5. The SSAc values are estimated for Ci-Cs using a LUT containing simulated diffuse radiation values. This LUT is generated with the input of 13 different SSAc values (between 0.8 and 1) and with the estimated $COT_{DSR}$ as input to libRadtran.

For St-As, in addition to the $COT_{DSR}$, we can also estimate the effective droplet radius ($r_{eff}$),

$$r_{eff} \approx \frac{3LWP}{2\rho_{lw}COT_{DSR}} \tag{4}$$

where LWP is the liquid water path, $COT_{DSR}$ the cloud optical thickness and $\rho_{lw}$ the density of liquid water (Stephens, 1994). The LWP is measured with a HATPRO instrument in Payerne and the $COT_{DSR}$ is estimated with the above-mentioned method.





## 4 Sensitivity Analysis

The aim of the method-related sensitivity analysis is to examine the robustness of the retrieved variables $COT_{DSR}$ and SSAc. These two variables are estimated from a LUT, which was generated using a radiative transfer model. In a first step, we examine the uncertainties as well as sensitivities of the RTM input parameters. For our analysis, we assume that all input variables are independent and uncorrelated and hence their influence on the radiation output can be estimated by varying each input parameter separately. To a large extend of the data set this assumption is true, however, for example the snow cases with high

albedo values have an influence on the sensitivity and thus on the uncertainty of COT and are thus not completely independent. In a second step, we multiply the standard uncertainties $u$ with the estimated sensitivities which results in an uncertainty value per parameter. In a third step, we calculate the combined uncertainties for the simulation of the radiation under the different sky conditions. These values are thereafter used to estimate the uncertainties of the COT and SSAc retrieval. The assumed uncertainties are Type B uncertainties which are uncertainties that are not based on statistical analysis but rather on uncertainties

specified in literature, experience or previous measurements (Guide to the Expression of Uncertainty in Measurement (GUM); BIPM (2008)). The uncertainties for the cloudy cases presented in Table 2 and Table 3 are estimated for the example cases for which COT values equal to 38 and 0.8 for St-As and Ci-Cs, respectively. The combined uncertainties for COT values were additionally calculated for other COT values between 10 and 100 (St-As) and 0 and 5 (Ci-Cs), but are not presented here. In summary, for stratus-altostratus, the larger the estimated COT value, the larger the absolute expanded combined uncertainty

value $U_c$. However, in relative uncertainties, independent of the estimated COT value, the uncertainty is around 18 %. For cirrus-cirrostratus the opposite applies, the $U_c$ in COT retrieval is 0.1 for all cases, independent of the COT value. A similar behaviour of the uncertainties of COT estimations are also presented in Serrano et al. (2014).

The estimated standard uncertainties $u$ for the specified input parameters in the libRadtran model are shown in Table 2 (second column). The standard uncertainty for IWV is taken from literature (Morland et al., 2006b).

The AOD data set consists of daily and monthly mean values, respectively. Therefore, the uncertainty $u$ for the AOD values under cloud-free conditions is estimated from the standard deviation comparing the used L3 MODIS AOD values with the measured PFR AOD data, where the mean difference is zero with a standard uncertainty of 0.07. For cloudy conditions, AOD can be measured by neither the PFRs nor by satellites. Assuming a rectangular distribution of the data, the uncertainty $u$ is calculated by dividing the half width of 95 % of the data set ($a$) by the root of three ($u = \frac{a}{\sqrt{3}}$). For AOD, under cloudy conditions,

$u$ was estimated with this formula for different seasons separately. The resulting uncertainty is 0.08, which is the standard uncertainty value used for AOD.

The uncertainty $u$ for albedo was calculated with the same equation, also taking into account 95 % of the data set and for different seasons separately, but neglecting the occasional snow events. The resulting $u$ value for albedo is 0.06.

The uncertainty of total column ozone is assumed to be 1 % (Levelt et al., 2018), which corresponds to an uncertainty of 3 -

4 DU.

The effective droplet and ice crystal radius values are assumed to be between 5 and 45 $\mu$m, also with a rectangular distribution. The sensitivities of the input parameters under cloudy conditions in Table 2 were calculated with COT values defined in the





libRadtran input file. Consequently, in the analysed cases, the LWC has a negligible influence on the calculation of the COT and is therefore not listed in Table 2. Also not listed are all variables that were not specifically defined in our analysis due to lack of available measurement data.

The uncertainties of the different input parameters under different sky conditions are used to calculate the combined uncertainty ($u_c$). The expanded combined uncertainty ($U_c$) is calculated thereafter representing 95 % of the data set assuming a normal distribution.

The global DSR under cloud-free conditions can be simulated with an expanded combined uncertainty of 2.4 %. Thus, this uncertainty is in a similar range as the instrument related shortwave radiation measurement uncertainty. Almost half of the estimated expanded combined uncertainty is caused by the uncertainty of the AOD (1.1 %) (Table 2, third column). The contribution to the uncertainty of the input parameters IWV and total column ozone is negligible.

For the simulation of the diffuse DSR under a St-As cloud with a COT value of 38, the parameter contributing most to the standard uncertainty of 7.3 % is the albedo with 6.9 %. The second largest contributor to the uncertainty budget is AOD with 1.7 % and hence represents a variable which in practice cannot be measured in the presence of a stratus cloud. The influence of the macrophysical properties, both cloud vertical thickness and CBH, on the DSR is negligible. The expanded combined model uncertainty (14.5 %) of the diffuse DSR under a stratus-altostratus cloud is thereafter used to estimate the uncertainty of the retrieved cloud optical thickness values shown in Table 3.

For the simulation of the direct radiation under a cirrus-cirrostratus cloud with COT equal to 0.8, the expanded combined uncertainty is, at 14.6 %, much larger than the model uncertainty of the diffuse radiation (4.6 %) under the same cloud conditions. Whereas for the direct radiation the dominant contributor to the expanded uncertainty is AOD (7.3 %), the main contributor to the expanded uncertainty of the diffuse radiation is the albedo (1.9 %).

The estimated model uncertainties presented in Table 2 are then used to calculate the expanded combined uncertainties of the COT retrieval (summarised in Table 3). The retrieval method of the COT values for St-As conditions presented here has a $U_c$ of 6.80 COT. The expanded combined uncertainty under Ci-Cs are for COT and SSAc 0.10 and 0.02, respectively.

## 5 Results

The optical thickness $\tau$ in the radiative transfer equation is a sum of optical thickness values of different atmospheric components (see Eq. 2). Therefore, to determine the optical thickness of clouds, the model is first validated for cloud-free values ($\tau_{clouds} = 0$), by assuring that including all other input parameters to the model leads to a reasonable calculation of the downward shortwave radiation.

### 5.1 Cloud-free

In the period January 1, 2013 to December 31, 2017, 13,240 cloud-free measurements on 379 days with SZA below 65° are available. The simulations of the global DSR for cloud-free cases show a very good agreement in comparison to the





**Table 2.** Uncertainty analysis for the radiation variables (glo: global, dir: direct, dif: diffuse) under different cloud conditions in absolute [$\mathrm{Wm}^{-2}$] and relative [%] numbers (in brackets). Cf: cloud-free, St-As: stratus-altostratus, Ci-Cs: cirrus-cirrostratus, u: standard uncertainty of the variables, $u_{xxx}$: u multiplied with the sensitivity value, $u_c$: combined standard uncertainty, $U_c$: expanded combined uncertainty (covering 95 % of the data set). The sensitivities were estimated with assumed COT values of 38 (for St-As) and 0.8 (for Ci-Cs). Estimated irradiances to calculate the relative numbers: Cf glo: 942.1 $\mathrm{Wm}^{-2}$, St-As dif: 156.2 $\mathrm{Wm}^{-2}$, Ci-Cs dir: 201.9 $\mathrm{Wm}^{-2}$, Ci-Cs dif: 442.5 $\mathrm{Wm}^{-2}$.

|  |  | Cf | St-As | Ci-Cs | Ci-Cs |
|---|---|---|---|---|---|
|  | u | $u_{glo}$ | $u_{dif}$ | $u_{dir}$ | $u_{dif}$ |
| AOD | 0.08 | 10.0 (1.1) | 3.5 (1.7) | 13.2 (7.3) | 1.4 (0.3) |
| IWV | 1 mm | 2.7 (0.3) | 0.5 (0.3) | 0.5 (0.3) | 1.1 (0.2) |
| Ozone | 4 DU | 0.2 (0.0) | 0.1 (0.0) | 0.0 (0.0) | 0.1 (0.0) |
| Albedo | 0.06 | 5.1 (0.5) | 14.2 (6.9) | - | 8.6 (1.9) |
| $r_{eff}$ | 11.55 $\mu$m | - | 3.2 (1.5) | 0.4 (0.2) | 3.8 (0.9) |
| vert. thick. | 0.78 km | - | 0.6 (0.3) | - | 0.1 (0.0) |
| CBH | 1 km | - | 1.3 (0.6) | - | 0.8 (0.2) |
| $u_c$ |  | 11.5 (1.2) | 15.0 (7.3) | 13.2 (7.3) | 9.6 (2.1) |
| $U_c$ |  | 23.0 (2.4) | 30.0 (14.6) | 26.4 (14.6) | 19.2 (4.2) |

**Table 3.** Uncertainty analysis for estimated COT values for stratus-altostratus (St-As) and cirrus-cirrostratus (Ci-Cs) and SSAc in absolute [$\mathrm{Wm}^{-2}$] and relative [%] numbers (in brackets). $U$: standard uncertainty of the variables, $U_{xxx}$: U multiplied by the sensitivity value, $U_c$: expanded combined uncertainty. The values were estimated with assumed COT values of 38 (for St-As) and 0.8 (for Ci-Cs).

|  |  | St-As | Ci-Cs | Ci-Cs |
|---|---|---|---|---|
|  | U | $U_{COT}$ | $U_{COT}$ | $U_{SSAc}$ |
| dif St-As meas | 3.1 (2.0) | 1.19 | - | - |
| dif St-As mod | 30.0 (14.6) | 6.69 | - | - |
| dir Ci-Cs meas | 3.9 (2.0) | - | 0.01 | - |
| dir Ci-Cs mod | 26.4 (14.6) | - | 0.10 | - |
| dif Ci-Cs meas | 7.8 (2.0) | - | - | 0.01 |
| dif Ci-Cs mod | 19.2 (4.2) | - | - | 0.02 |
| $U_c$ |  | 6.80 | 0.10 | 0.02 |





measurements. The absolute and relative mean difference (absolute difference divided by the measured value) between the modelled and the measured DSR is 5.71 $\mathrm{Wm}^{-2} \pm 11.91 \mathrm{Wm}^{-2}$ and 0.93 % $\pm$ 2.08 %, respectively. Thus the model is slightly overestimating the DSR measurement but the agreement is within measurement uncertainty of the instrument (2 %) (Vuilleumier et al., 2014) as well as within the estimated expanded combined uncertainty of 2.4 % (discussed in Section 4). The good agreement between the modelled and the measured global DSR is also demonstrated in the high correlation

coefficient (r=0.996). There is no temporal trend in the difference between the modelled and the measured DSR throughout the whole time period, which confirms the stability of the instrument as already discussed in Vuilleumier et al. (2014). Analysis of the difference between the simulated and the measured DSR values per day of year shows no seasonal dependence of the agreement. Consequently, we can conclude that the simulation of DSR under cloud-free conditions is excellent.

Comparing separately the two components of the total DSR (direct and diffuse) shows that in general, the direct radiation has a

larger correlation (r=0.98) between measurements and simulations than the diffuse component (r=0.73). The better agreement of the direct radiation is also reflected in the relative mean difference (modelled minus measured divided by measured) of -0.21 % $\pm$ 6.16 % in comparison to the relative mean difference of the diffuse radiation of 10.04 % $\pm$ 21.49 %. Figure 2 shows the distribution of the absolute differences between the modelled and the measured direct (top) and diffuse (bottom) radiation. In the mean, the model is slightly underestimating the measured direct radiation by -3.13 $\mathrm{Wm}^{-2} \pm 28.62 \mathrm{Wm}^{-2}$ and the

modelled diffuse radiation is slightly overestimating the measurement by 7.66 $\mathrm{Wm}^{-2} \pm 20.16 \mathrm{Wm}^{-2}$. The small difference between the modelled and measured direct radiation can for example be explained by uncertainties due to differences in the forward scattering due to different fields of view of the instrument and the model (Blanc et al., 2014) or by differences in the actual and RTM used extraterrestrial solar irradiance. However, the good agreement in the direct radiation confirms the proper use of the RTM AOD inputs. Part of the larger difference of the diffuse radiation can be explained by the use of default

values for the atmospheric profile instead of radiosonde data. However, as discussed in Section 3.1.1 this difference is small. Adjusting the aerosol single scatter albedo per case also decreases the difference in the diffuse radiation. However, due to the lack of aerosol SSA measurements, no further improvement in such deviations is possible in the current study. In summary, we found a similar agreement in the global and direct shortwave radiation as other groups in the past (e.g. Kato et al., 1997; Michalsky et al., 2006; Nowak et al., 2008a; Wang et al., 2009; Ruiz-Arias et al., 2013; Dolinar et al., 2016).

Consequently, because the simulation of DSR under cloud-free conditions achieved an agreement with the measured DSR within measurement and model uncertainty, we assume that all input parameters in Equation 2, except the COT, are well-defined. Subsequently, a similar model layout is used to simulate the DSR under cloudy conditions.

## 5.2 COT, SSAc and $r_{eff}$ estimations

### 5.2.1 Stratus-Altostratus

The data set of St-As consists of 3,724 measurements collected on 312 days. In cases of thick, low-level water clouds, the direct component of the radiation is less than 1 $\mathrm{Wm}^{-2}$. Thus, for these cases the global DSR is nearly only diffuse radiation due to multiple scattering at the cloud droplets as well as absorption and scattering above and below the cloud. In the case



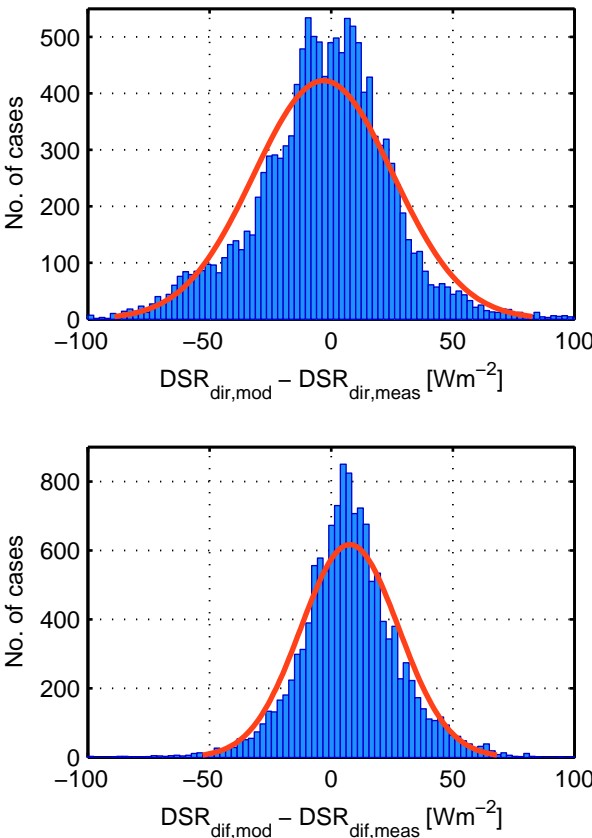

**Figure 2.** Distribution of the residuals of the differences between modelled and measured global downward shortwave radiation for cloud-free cases for the direct (top) and the diffuse (bottom) component in Payerne.

of low-level clouds, the most relevant optical property for the simulation of cloudy conditions is the COT. The default SSAc value used for the simulation of radiation can be a source of uncertainty in the COT determination, however Rawlins and Foot
(1990) pointed out that it is an input parameter of minor importance for this cloud class.

The resulting distribution of the estimated $COT_{DSR}$ values for our data set in Payerne is shown in Figure 3. The arithmetic mean $COT_{DSR}$ value retrieved from our analysis is 38.5 ± 20.6. Considering a lognormal distribution, the geometric mean of 33.81 with a geometric standard deviation of 1.67 represents a range in $COT_{DSR}$ values between 20.26 and 56.43. The variability of $COT_{DSR}$ values is much larger than the expanded combined uncertainty $U_c$ of the COT retrieval. Thus, the large
variability in $COT_{DSR}$ values for St-As cases in Payerne is reflecting the inhomogeneity of these clouds and is not due to the uncertainty in the retrieval method. Ninety-five percent of the $COT_{DSR}$ values for the St-As data set are between 11.9 and 91.5. This finding of a minimum $COT_{DSR}$ value of 11.9 agrees with the findings of Bohren et al. (1995) stating that the direct





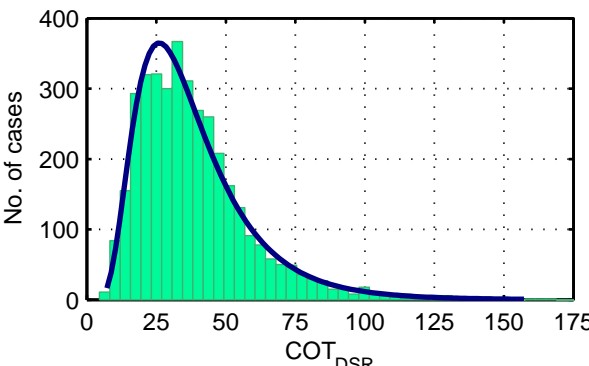

**Figure 3.** Distribution of COT retrieved from DSR measurements ($COT_{DSR}$) for stratus-altostratus cases in Payerne. The geometric mean $COT_{DSR}$ value is 33.81 with a geometric standard deviation of 1.67.

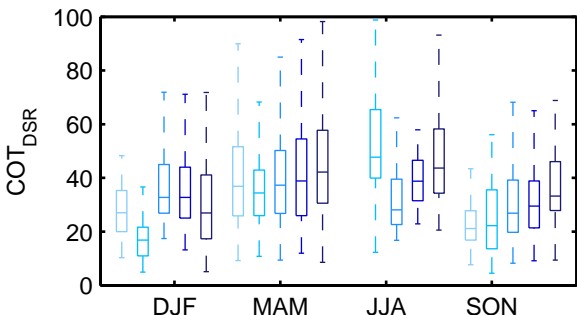

**Figure 4.** Distribution of the $COT_{DSR}$ values per season (DJF: winter, MAM: spring, JJA: summer, SON: autumn) and years (light to dark blue: 2013 to 2017) of St-As in Payerne. No data in JJA 2013.

shortwave radiation is blocked if COT is larger than 10.

Figure 4 shows the distribution of the $COT_{DSR}$ values in different seasons and years. The boxplots show the median, the interquartile range and the 95 % intervals of the $COT_{DSR}$ values. It demonstrates, that the $COT_{DSR}$ values are in general higher in spring (MAM) and summer (JJA), than in autumn (SON) and winter (DJF). This finding is consistent with a study presenting the COT distribution over the seasons at different stations in China (Li et al., 2019). In spring and autumn it seems that the $COT_{DSR}$ values increase with time. But the statistic is too small to draw any conclusions about a trend.

Figure 5 shows the fluctuation of $COT_{DSR}$ (blue) and LWP (red) within a few hours on March 15, 2015 during St-As
conditions. Within a short time period (less than 40 minutes), the $COT_{DSR}$ decreases about 20 - 30 units (in Figure 5 between 10:15 and 10:45 UTC). The visual checking of the corresponding images confirms nicely the dissipation of the thick cloud layer to a thinner one. This dissolving of the cloud layer in Payerne around local noon also matches the typical meteorological





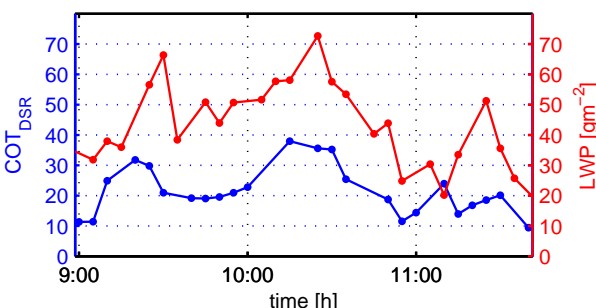

**Figure 5.** Time series of the $COT_{DSR}$ (blue) and LWP (red) during St-As conditions in Payerne on March 15, 2015.

situation of the location. The change of $COT_{DSR}$ also correlates with independent measurements of LWP from a HATPRO instrument: the smaller the $COT_{DSR}$, also the smaller the LWP value. The short-term changes of $COT_{DSR}$ values (two

consecutive measurements 5 min apart) of less than 5 are within the $COT_{DSR}$ retrieval uncertainty, which is discussed in Section 4.

The $COT_{DSR}$ values of St-As are thereafter used to estimate $r_{eff}$ using Equation 4. For this estimation all LWP data with values greater than 400 $gm^{-2}$ are neglected due to the presence of rain, as well as all values below 30 $gm^{-2}$ because this threshold corresponds to cloud-free conditions (Löhnert and Crewell, 2003). The determined mean $r_{eff}$ for our St-As data set

is 7.0 $\mu$m $\pm$ 4.6 $\mu$m. The mean value agrees with the value presented in Hess et al. (1998) for continental stratus clouds. The 2.5th and 97.5th percentiles of the determined $r_{eff}$ are 2.1 $\mu$m and 20.4 $\mu$m, respectively.

### 5.2.2 Cirrus-Cirrostratus

A similar analysis to the one for St-As is also performed for the high-level cloud class Ci-Cs. As already mentioned in Section 2.2, the data set of Ci-Cs consists of 206 measurements on 48 days. The distribution of the $COT_{DSR}$ values estimated

from the direct shortwave irradiance is shown in Figure 6. The mean $COT_{DSR}$ is 0.75 $\pm$ 0.26 and 95 % of the $COT_{DSR}$ values vary between 0.32 and 1.40 and are thus in a similar range as, for example, presented in Giannakaki et al. (2007) and Hong and Liu (2015). Also, the expanded combined uncertainty of the $COT_{DSR}$ retrieval method under Ci-Cs conditions (0.10), is much smaller than the one sigma $COT_{DSR}$ variability (0.26). The latter is therefore also reflecting the large variability in the $COT_{DSR}$ values in the Ci-Cs data set.

The $COT_{DSR}$ values retrieved are used as input to libRadtran in order to estimate the SSAc values for Ci-Cs by minimising the diffuse radiation residuals. The mean SSAc value retrieved is 0.92 $\pm$ 0.04 (Figure 7) and therefore slightly larger than the libRadtran default value of 0.87 (Key et al., 2002). 95 % of the SSAc data are between 0.84 and 0.99. Therefore, we can conclude that the SSAc values defined by Key et al. (2002) are mostly underestimating the extinction by scattering for the cirrus-cirrostratus data set in Payerne.

The SSAc under Ci-Cs conditions can be determined with an uncertainty of 0.02 which is smaller than the one sigma variabil-





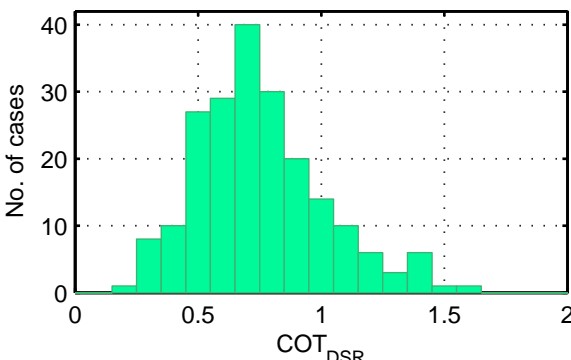

**Figure 6.** Distribution of $COT_{DSR}$ retrieved from direct DSR measurements for cirrus-cirrostratus cases in Payerne. The mean $COT_{DSR}$ value is $0.75 \pm 0.26$.

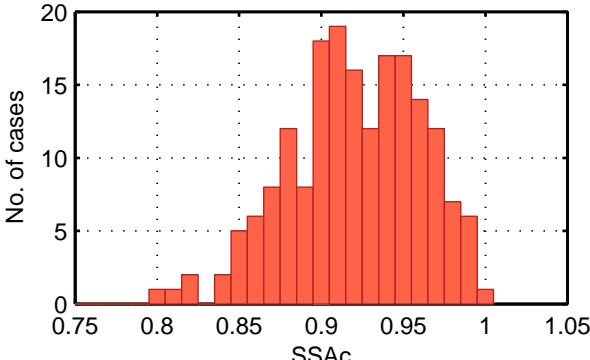

**Figure 7.** Distribution of cloud single scattering albedo (SSAc) retrieved from diffuse DSR measurements for cirrus-cirrostratus cases in Payerne. The mean SSAc value is $0.92 \pm 0.04$.

ity of 0.04. Thus, the variability in the results for SSAc is larger than the model uncertainty and confirms the importance of accurate knowledge of the SSAc values for high-level clouds.



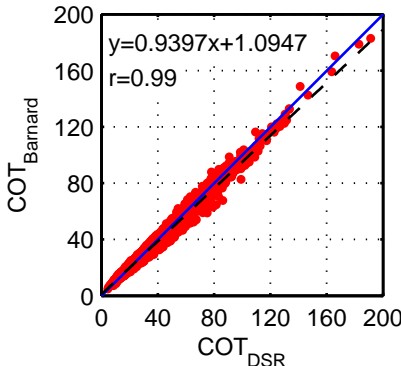

**Figure 8.** Correlation between COT values retrieved from DSR measurements ($COT_{DSR}$) and from the equation presented in Barnard and Long (2004) ($COT_{Barnard}$) for St-As in Payerne.

## 6  Comparison $COT_{DSR}$ with independent data sets

### 6.1  Barnard and Long equation

Our retrieved $COT_{DSR}$ values for St-As are compared to COT values estimated by applying the empirical equation by Barnard and Long (2004),

$$\tau_c = exp(2.15 + A + 1.91 * arctanh(1 - 1.74 * \frac{D}{C\mu_0^{\frac{1}{4}}}))  \tag{5}$$

where $\tau_c$ is the cloud optical thickness (here $COT_{Barnard}$), $A$ is the albedo, $D$ the measured broadband diffuse radiation, $C$ a clear-sky model value and $\mu_0$ the airmass. In the current study the clear-sky model values are estimated according to Aebi et al. (2017). The correlation between the $COT_{DSR}$ and $COT_{Barnard}$ is very high (r=0.99) (Figure 8). The mean COT difference between these two retrieval methods is -1.20 $\pm$ 2.73, showing a slight underestimation of $COT_{Barnard}$. However this difference is within the model uncertainty.

The COT estimation formula presented in Barnard and Long (2004) is only valid for thick clouds with COT values larger than 10. Consequently, this formula cannot be applied to Ci-Cs cases because the diffuse radiation is not the correct component for estimation of the COT.

### 6.2  MODIS

The $COT_{DSR}$ values are also compared with L2 C6.1 COT values from MODIS Aqua ($COT_{MODIS}$). The comparison is performed for a subset of the St-As data set, taking into account the overpass time of the MODIS satellite. The analysis is done for MODIS grid points of 3×3 km, 5×5 km and 7×7 km above Payerne. Considering the mean $COT_{DSR}$ value from data $\pm$ 30 min around the overpass times of the satellite and the highest spatial resolution results in a matching in 169 cases. At 37.59 (geometric standard deviation 1.68), the geometric mean of $COT_{DSR}$ for this subset is much higher than the geometric





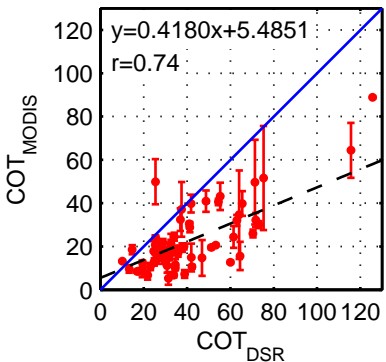

**Figure 9.** Correlation between COT values retrieved from DSR measurements ($COT_{DSR}$) in Payerne and from MODIS level-2 data ($COT_{MODIS}$) in a grid point of $3 \times 3$ km for St-As.

mean and standard deviation of $COT_{MODIS}$ (17.35 and 1.90, respectively).

Considering only the $COT_{DSR}$ values which have an exact matching in time with the $COT_{MODIS}$ measurements decreases
the subset to 60 COT measurements, but does not decrease the difference between $COT_{DSR}$ and $COT_{MODIS}$. The geometric mean, geometric standard deviation and 2.5th and 97.5th percentiles for $COT_{DSR}$ and $COT_{MODIS}$ with the different satellite resolutions are shown in Table 4. The $COT_{DSR}$ is higher than the value mentioned in Section 5.2.1 because here only a subset of 60 measurements is taken into account. It is noteworthy that the difference in the mean of $COT_{MODIS}$ with different resolutions is small. However, at around 18, the difference in the geometrical mean between $COT_{DSR}$ and $COT_{MODIS}$
is rather high. The correlation between $COT_{DSR}$ and $COT_{MODIS}$ for the 3×3 km resolution is r=0.74 (Figure 9). Li et al. (2019) found similar correlation coefficients for stations in China for instantaneous matching of COT data from MODIS and radiometers. In their study the $COT_{MODIS}$ values are in general also lower than the ground-based COT values. The satellite analysis may only take into account the highest cloud layer, while the values derived from DSR take into account all layers, even though the camera did not allow identification of cases when multiple cloud layers were present. Another explanation
might be the slight difference in the wavelength (Baum et al., 2014).

We also used the $COT_{MODIS}$ and $r_{eff,MODIS}$ (also L2) and a grid of 3×3 km to calculate the DSR with libRadtran. This analysis results in a mean underestimation of $DSR_{MODIS}$ of 88 Wm$^{-2}$ in comparison to the measured DSR during St-As conditions in Payerne.

Other studies (e.g. Painemal and Zuidema, 2011; McHardy et al., 2018) show a better agreement between ground- and satellite-
based COT values, but mainly for averaged data over a longer time period (for example monthly means). The sample of 60 data points is too small to calculate a monthly mean COT.

Comparing the $COT_{DSR}$ and $COT_{MODIS}$ values for Ci-Cs shows only three matches in time. For these three situations, the $COT_{MODIS}$ is larger than $COT_{DSR}$. But the data set is too small to draw any conclusions from this comparison.





**Table 4.** Geometric mean, geometric standard deviation and 2.5th and 97.5th percentile values of COT retrieved from ground-based broadband shortwave radiation ($COT_{DSR}$) and from MODIS L2 data with different spatial resolutions (3×3 km, 5×5 km and 7×7 km) above Payerne.

| COT | Geom. Mean | Geom. Std. | 2.5th | 97.5th |
|---|---|---|---|---|
| DSR | 37.97 | 1.70 | 13.25 | 125.68 |
| 3×3 km | 19.60 | 1.80 | 6.73 | 64.39 |
| 5×5 km | 19.92 | 1.75 | 8.53 | 65.40 |
| 7×7 km | 20.20 | 1.74 | 8.66 | 64.28 |

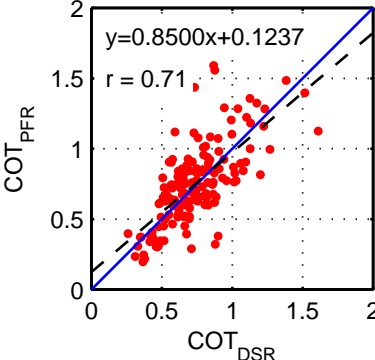

**Figure 10.** Correlation between COT values retrieved from DSR measurements ($COT_{DSR}$) and COT values retrieved from PFR measurements at $\lambda = 412$ nm ($COT_{PFR}$) for Ci-Cs in Payerne.

## 6.3 PFR

The $COT_{DSR}$ derived for the cirrus-cirrostratus cases are compared with the cloud optical thickness values derived from measurements of direct solar irradiance obtained from four collocated PFR sunphotometers measuring at 16 wavelengths between 305 and 1,024 nm ($COT_{PFR}$). The COT values are retrieved at the different channels of the instruments and corrected by the corresponding AOD values for the corresponding day. It is difficult to estimate the effective wavelength that corresponds to the $COT_{DSR}$ values derived from broadband measurements. As an example, Figure 10 shows a scatter plot of the $COT_{PFR}$ derived at 412 nm versus $COT_{DSR}$. The correlation of the COT between these two independent methods is 0.71. The slightly higher values of $COT_{PFR}$ relative to $COT_{DSR}$ might result from the different spectral regions used to retrieve the cloud optical thickness: the 412 nm channel for the PFR and the complete shortwave spectrum for $COT_{DSR}$. The correlation between $COT_{DSR}$ and $COT_{PFR}$ at 500 $nm$ is slightly lower (r=0.60). A slight dependence of the wavelength on the retrieved COT values is also confirmed by the analysis of the COT values retrieved at other wavelengths of the PFRs. Another explanation for the discrepancy might be the enhanced forward scattering entering the field of view of the instrument, which causes an overestimation of the measured direct shortwave radiation compared to the modelled one (Blanc et al., 2014). This fact results in an underestimation of $COT_{PFR}$ of Ci-Cs clouds.


## 7 Summary and Conclusions

The current study presents a method to retrieve COT, SSAc and $r_{eff}$ values for the two cloud classes stratus-altostratus and cirrus-cirrostratus by combining broadband solar shortwave radiation (total as well as the direct and diffuse components) measurements with a radiative transfer model. The study is performed with radiation data from the BSRN station, Payerne, Switzerland, which can be seen as a reference station for radiation measurements and thus our method can also be applied at other stations. In total, more than 3,000 St-As measurements and 206 Ci-Cs measurements collocated in the time period

January 1, 2013 to December 31, 2017 and in situations with a SZA lower than 65° are analysed.

In order to test the model-measurement combination performance, in a first step more than 12,000 cloud-free measurements were analysed. With a relative mean difference of 0.93 % ± 2.08 %, the simulated cloud-free global DSR is in agreement with the measured global DSR within instrument uncertainty. The sensitivity analysis shows an expanded model uncertainty (covering 95 % of the data set) of DSR retrieval of less than 2.5 % and thus the difference is also within the model uncertainty.

Ninety-five percent of the estimated St-As $COT_{DSR}$ values are between 11.9 and 91.5 with a geometric mean and geometric standard deviation of 33.81 and 1.67. The $COT_{DSR}$ values are higher in spring and summer than in autumn and winter. These estimated $COT_{DSR}$ values are in very good agreement with the $COT_{Barnard}$ values estimated using the empirical equation of Barnard and Long (2004). The mean difference in the COT values between these two methods is -1.20 ± 2.73, which is within model uncertainty. However, for a subset of the St-As data set, $COT_{MODIS}$ with a resolution of 3×3 km is clearly

underestimating our determined $COT_{DSR}$ values. Using $COT_{MODIS}$ and $r_{eff}$ from MODIS to estimate DSR results in a mean underestimation of the global irradiance of more than 50 % of the measured DSR values in Payerne. Changing the spatial resolution and/or the matching in time does not result in a smaller difference in the mean COT. However, these large discrepancies cannot be explained at present, but were also shown in other studies (e.g. Li et al., 2019).

The 2.5th and 97.5th percentiles in $r_{eff}$ under St-As conditions in Payerne are 2.1 $\mu$m and 20.4 $\mu$m, respectively and thus are

comparable to values presented in other studies (e.g. Hess et al., 1998).

The retrieved mean $COT_{DSR}$ value from direct broadband shortwave radiation under Ci-Cs conditions in Payerne is 0.83 ± 0.27 and thus in a similar range as described in other studies (e.g. Qiu, 2006; Giannakaki et al., 2007; Hong and Liu, 2015). The comparison of the $COT_{DSR}$ and the $COT_{PFR}$ values retrieved from PFRs shows correlation coefficients at r=0.60 (500 $nm$) and r=0.71 (412 $nm$). The retrieved mean cloud single scattering albedo value for Ci-Cs is 0.91 ± 0.04.

It has been demonstrated, that with this method COT, SSAc and $r_{eff}$ can be estimated from state-of-the-art data sets in Payerne. The same method could also be applied at other BSRN stations in order to validate the method. In the case of similar results in the $COT_{DSR}$ estimation, a long-term data set in cloud properties could be produced and could be of use to increase the availability of cloud optical parameters for e.g. climate models.

An extension of this study would be to perform a radiative closure study for longwave radiation for a similar data set. This

analysis would be an extension of the study presented by Wacker et al. (2011) which describes a longwave closure study for well-defined stratus nebulosus cases in Payerne. This future analysis is important in order to analyse the effect of cloud micro-





physical properties on longwave radiation as well and to develop thereafter a more complete picture of the influence of cloud parameters on the surface radiation budget.

*Data availability.* Data are available from the corresponding author on request.

*Competing interests.* The authors declare that they have no conflict of interest.

*Acknowledgements.* This research was carried out within the framework of the project *A Comprehensive Radiation Flux Assessment (CRUX)* funded by MeteoSwiss. We would like to thank Giovanni Martucci from MeteoSwiss for providing us with the PFR data. We also acknowledge Maxime Hervo from MeteoSwiss who provided the LWP data. We acknowledge the tropospheric emission monitoring internet service web page www.temis.nl for providing the OMI total column ozone data. MODIS data used herein were produced with the Giovanni online
data system, developed and maintained by the NASA GES DISC: https://giovanni.gsfc.nasa.gov/giovanni/doc/UsersManualworkingdocument.docx.html.



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
