# Peer review of "Estimation of cloud optical thickness, single scattering albedo and effective droplet radius using a shortwave radiative closure study in Payerne"

_Atmospheric Measurement Techniques, 2019_

## Referee Comment (RC1) · Anonymous Referee #1 · 1 Dec 2019

Review of the study entitled "Estimation of cloud optical thickness, single scattering albedo and effective droplet radius using a shortwave radiative closure study in Payerne" by Aebi et al.

The study presents a method for estimating the cloud optical thickness, single scattering albedo and effective droplet radius from downward shortwave radiation simulations and measurements during days with low clouds (stratus-altostratus) and high clouds (cirrus-cirrostratus) in Payerne, Switzerland in the period 2013-2017. The authors have done a good job to describe their analysis, and demonstrate their findings in a good

way. Results from their method are tested against other methods and good correlations are revealed. The study is suitable for publication in AMT, the manuscript is well written, the findings are well described and in general I find a good paper. I recommend publication after a few clarifications.

Comment 1: Table 1: The table refers to 739 out of 1827 days during the period 2013-2017. Maybe it escaped my sight but I couldn't find out what happened with the remaining days. Can you explain what happens with the remaining 3/5 of the period, e.g. no available sky camera measurements, different cloud types than the ones investigated?

Comment 2: I am confused with the use of terms COT and COT_DSR. From the abstract I understand that COT is the cloud optical thickness calculated from modelled downward shortwave radiation, and that COT_DSR is the cloud optical thickness derived from measured downward shortwave radiation. However in section 3.2, I read that the total DSR and its components, direct and diffuse radiation, are derived from libRadtran, and that the lookup tables, used to estimate the COT_DSR, contain simulated radiation values. I cannot figure out how the ground-based radiation measurements are used to derive the COT_DSR. Please clarify.

Comment 3: In the same motif. Lines 234-235: What is the 'effective COT_DSR'? Do you mean that a modelled COT is used as input to derive the measured COT_DSR? Line 251: It reads 'COT_DSR the cloud optical thickness'. Is this derived from radiation measurements? Line 256: It reads 'These two variables are estimated from a LUT, which was generated using a radiative transfer model'. So, is COT_DSR derived from simulated radiation values and not from measured ones?

Comment 4: Line 310: it reads '6.80 COT'. Is it '6.80 of COT' or is it just '6.80' and COT is a typo error?

---

## Referee Comment (RC2) · Josep Calbó (Referee) · 3 Dec 2019

This Discussion paper addresses a matter that is of current interest: the determination of optical characteristics of clouds based on measurements and observations from the surface. Specifically, the authors suggest a new method to determine cloud optical thickness (COT) from the measurements of shortwave (solar) irradiances at the ground. The method is suitable for stratiform cloudiness, either low thick clouds (St, As), when diffuse irradiance is used; or thin high clouds (Ci, Cs) when direct irradiance is of use. In both cases, besides the COT another cloud characteristic can be derived: droplet

effective radius in one case; single scattering albedo in the other case.

Research is correctly designed, and the data used to develop and apply de method is also adequate. Several comparisons with previous methods that estimated COT are presented, which is also a value of this study. I would say that the method is somewhat brute-force (as many simulations are performed with the radiative transfer model for each particular measurement), but this is not a problem as the computational cost is relatively.

The only general comment, which is not really a criticism but a question that the authors may address either in the introduction or, more precisely, in the discussion of the results, regards the practical utility of this method. If I understood correctly, estimates of COT by the new method essentially match estimates with Barnard-Long (2004) expression. Being the latter much simpler that the former, in what conditions the new method is useful?

Some other minor comments and suggestions follow:

- Introduction. The work Matamoros, S., González, J.-A., & Calbó, J. (2011). A Simple Method to Retrieve Cloud Properties from Atmospheric Transmittance and Liquid Water Column Measurements. Journal of Applied Meteorology and Climatology, 50(2), 283–295. https://doi.org/10.1175/2010JAMC2394.1 could be of interest for the authors, as it presents also a simple method based on the measurements of transmittance at 415 nm.

- Line 75-77. "[. . .] can provide the limits of the minimum possible agreement[. . .]" Does this mean simply "the best possible agreement"?

- Section 2.1. Although pretty obvious, should you mention that you use a sun-tracker for diffuse and direct irradiance measurements? And, what does "homogenized" mean in this context (this is a term usually employed in a climatic, that is, long-term series, context)?

- Fig. 1. You should explain if the data shown is only for cloudless days (as I think is the case for AOD), or for all days.

- Section 2.2., lines 136-137, it is not needed to repeat the period included in the analysis.

- Line 189, Kokhanovsky is good reference, but not the only source where optical properties of clouds are explained. You could add "for example".

- Why do you need a section 3.1.1 if there is no section 3.1.2?

- Starting in section 3.1 but then across all the rest of the paper. Be cautious with the use of significant figures in the numbers you give. Many times you give four significant figures, but only one (or two) are really significant. For example, 53.54 % $\pm$ 20.92 should be 53 % $\pm$ 21; 5.71 Wm$-2$ $\pm$ 11.91 should be 6 $\pm$ 12, and so on.

- Lines 233-234. The sentence "As ice habit the default solid column is included" sounds awkward to me.

- Section 3.2. I suggest reordering the description of these methods. First, all methods regarding St-As, that is COT and reff; then, methods regarding Ci-Cs, that is COT and SSA.

- Line 522. In the first moment, it is not clear if DSR refers here to global (or total) irradiance. Later, the reader infers it, as results for direct and diffuse are commented. But I suggest indicating always which irradiance the authors refer to. In addition, I encourage consistency regarding the use of "total" and "global".

- Final comment in section 5.1. It should be discussed the fact that AOD cannot be measured under cloudy conditions. So, strictly speaking, AOD data are not introduced the same way under cloud-free and under cloudy conditions.

- Section 6.2. I don't understand one result commented here. If MODISs underestimates COT, these lower COT should result in an overestimation of DSR.

- Conclusions. Your results seem to indicate that COT_modis are not correct, at least not correct under conditions analyzed in the present study. I suggest adding this conclusion.

Technical matters:

- Please check acronyms and their definitions. For example, DSR is defined in the abstract, but not before its first use in the main text. Or, SSA is defined in line 187 but it has been used many times before.

- Table 3. COT and SSA are dimensionless variables; they don't have Wm-2 units.

---

## Author Comment (AC1) · 7 Jan 2020

**Reply to comments by Anonymous Referee #1**

on the manuscript "Estimation of cloud optical thickness, single scattering albedo and effective droplet radius using shortwave radiative closure study in Payerne" by Aebi et al., submitted to Atmospheric Measurement Techniques.

We thank the referee for the constructive comments that contributed to the improvement of the manuscript. Detailed answers to the comments are given below (bold: referee comment, regular font: author's response, italic: changes in the manuscript).

**The study presents a method for estimating the cloud optical thickness, single scattering albedo and effective droplet radius from downward shortwave radiation simulations and measurements during days with low clouds (stratus-altostratus) and high clouds (cirrus-cirrostratus) in Payerne, Switzerland in the period 2013-2017. The authors have done a good job to describe their analysis, and demonstrate their findings in a good way. Results from their method are tested against other methods and good correlations are revealed. The study is suitable for publication in AMT, the manuscript is well written, the findings are well described and in general I find a good paper. I recommend publication after a few clarifications.**

**Comment 1: Table 1: The table refers to 739 out of 1827 days during the period 2013-2017. Maybe it escaped my sight but I couldn't find out what happened with the remaining days. Can you explain what happens with the remaining 3/5 of the period, e.g. no available sky camera measurements, different cloud types than the ones investigated?**

The parameter "number of days" defines the number of days during which the total number of measurements of one specific sky condition is found, e.g. for Ci-Cs, the 206 measurements are found at 48 different days and for Cf, the 13,240 measurements are found at 379 different days. It is also possible that at the same day cloud-free and Ci-Cs measurements are available for example. At all other days, there was either another cloud class present than the ones considered, or the cloud fraction criterion was not fulfilled.

To clarify, we changed p. 6, l. 151f:

*Table 1 summarises the number of measurements in total and the number of days during which they are found, the cloud fractions considered and the occurrences throughout the year per cloud class separately.*

**Comment 2: I am confused with the use of terms COT and COT_DSR. From the abstract I understand that COT is the cloud optical thickness calculated from modelled downward shortwave radiation, and that COT_DSR is the cloud optical thickness derived from measured downward shortwave radiation. However in section 3.2, I read that the total DSR and its components, direct and diffuse radiation, are derived from libRadtran, and that the lookup tables, used to estimate the COT_DSR, contain simulated radiation values. I cannot figure out how the ground-based radiation measurements are used to derive the COT_DSR. Please clarify.**

The COT$_{DSR}$ values are retrieved by combining DSR measurements with simulations of a RTM. To make it more clear, we rewrote Section 3.2. Additionally, we slightly changed the descriptions of the retrieval of COT$_{DSR}$ at various places. One example of the changes is:

p. 3, l. 74ff:
*In the current study we estimate cloud optical thickness for stratus-altostratus (St-As) and cirrus-cirrostratus (Ci-Cs) using broadband shortwave radiation measurements, a RTM and ancillary ground- and satellite-based data from the BSRN station in Payerne, Switzerland, by performing a radiative closure study. This allows determining COT by minimization of the difference between modelled and measured DSR values. The COT values determined with this method are abbreviated with COT$_{DSR}$.*

Throughout the manuscript we also used more consistently the term COT$_{DSR}$ and added some more times COT$_{MODIS}$, COT$_{Barnard}$ or COT$_{PFR}$ instead of only using COT. We hope that this leads to less confusion.

**Comment 3: In the same motif. Lines 234-235: What is the 'effective COT_DSR'? Do you mean that a modelled COT is used as input to derive the measured COT_DSR? Line 251: It reads 'COT_DSR the cloud optical thickness'. Is this derived from radiation measurements? Line 256: It reads 'These two variables are estimated from a LUT, which was generated using a radiative transfer model'. So, is COT_DSR derived from simulated radiation values and not from measured ones?**

l. 234-234: we deleted the term effective.

Otherwise, we changed the description of the COT$_{DSR}$ retrieval (Section 3.2, p. 9, l. 244ff) to make the method more clear (see also the answer to comment 2).

**Comment 4: Line 310: it reads '6.80 COT'. Is it '6.80 of COT' or is it just '6.80' and COT is a typo error?**

Thanks for this comment, it was indeed a typo and we changed the sentence to:

p. 11, l. 314f:
The retrieval *method of the COT values for St-As conditions presented here has a $U_c$ of 6.8.*

---

## Author Comment (AC2) · 7 Jan 2020

**Reply to comments by J. Calbó**

on the manuscript "Estimation of cloud optical thickness, single scattering albedo and effective droplet radius using shortwave radiative closure study in Payerne" by Aebi et al., submitted to Atmospheric Measurement Techniques.

We thank the referee J. Calbó for the constructive comments that contributed to the improvement of the manuscript. Detailed answers to the comments are given below (bold: referee comment, regular font: author's response, italic: changes in the manuscript).

**This Discussion paper addresses a matter that is of current interest: the determination of optical characteristics of clouds based on measurements and observations from the surface. Specifically, the authors suggest a new method to determine cloud optical thickness (COT) from the measurements of shortwave (solar) irradiances at the ground. The method is suitable for stratiform cloudiness, either low thick clouds (St, As), when diffuse irradiance is used; or thin high clouds (Ci, Cs) when direct irradiance is of use. In both cases, besides the COT another cloud characteristic can be derived: droplet effective radius in one case; single scattering albedo in the other case.**

**Research is correctly designed, and the data used to develop and apply de method is also adequate. Several comparisons with previous methods that estimated COT are presented, which is also a value of this study. I would say that the method is somewhat brute-force (as many simulations are performed with the radiative transfer model for each particular measurement), but this is not a problem as the computational cost is relatively.**

**The only general comment, which is not really a criticism but a question that the authors may address either in the introduction or, more precisely, in the discussion of the results, regards the practical utility of this method. If I understood correctly, estimates of COT by the new method essentially match estimates with Barnard-Long (2004) expression. Being the latter much simpler that the former, in what conditions the new method is useful?**

We agree, that the empirical equation from Barnard and Long (2004) is less time consuming than our method. However, as mentioned in the introduction, this empirical equation (or also the one shown in Matamoros et al., 2011) is only applicable for low-level clouds with COT larger than 10. In contrast, our method can be applied to clouds independent of their cloud level, which is one of the advantages of our method. The sentences mentioning this advantage are shown on:

p. 3, l. 54ff.:
*It has been proven that both empirical equations can be used for homogeneous low-level (COT>10), but not for high-level clouds. The aim of our study is to use a method which is based on a radiative closure study and which allows the determination of COT independent of the cloud level.*

**Some other minor comments and suggestions follow:**
**- Introduction. The work Matamoros, S., González, J.-A., & Calbó, J. (2011). A Simple Method to Retrieve Cloud Properties from Atmospheric Transmittance and Liquid Water Column**

Measurements. Journal of Applied Meteorology and Climatology, 50(2), 283–295. https://doi.org/10.1175/2010JAMC2394.1 could be of interest for the authors, as it presents also a simple method based on the measurements of transmittance at 415 nm.

Thanks for mentioning this paper, we added the following sentence in the introduction:

p. 2, l. 53f:
*Matamoros et al., 2010 presented another empirical equation to estimate COT from the atmospheric transmittance at 415 nm, SZA, surface albedo, $r_{eff}$ and aerosol optical depth.*

- Line 75-77. "[…] can provide the limits of the minimum possible agreement[…]" Does this mean simply "the best possible agreement"?

Thanks for this comment, we changed it accordingly:

p.3, l. 81ff:
*"… can provide the limits of the best possible agreement among modelled and measured …"*

- Section 2.1. Although pretty obvious, should you mention that you use a sun-tracker for diffuse and direct irradiance measurements? And, what does "homogenized" mean in this context (this is a term usually employed in a climatic, that is, long-term series, context)?

We added the sun-tracker:

p. 4, l. 101f:
*The diffuse and direct radiation values are measured with a CMP22 pyranometer and a CHP1 pyrheliometer, respectively, both installed on a sun tracker.*

The homogenization of the radiation data has been done according to the method described in Vuilleumier et al., 2014. This homogenization has not only been done for the five years data set we used, but for a longer time period. We added the aforementioned reference to our manuscript:

p. 4, l. 102ff:
*All radiation data are corrected for the thermal offset (Philipona, 2002), homogenised (Vuilleumier et al., 2014) and are available in a temporal resolution of 1 minute.*

- Fig. 1. You should explain if the data shown is only for cloudless days (as I think is the case for AOD), or for all days.

The data that are shown in Figure 1 are the values that are used as input to the RTM per day. They are either daily mean AOD values or if not available then it is a monthly mean value (calculated as explained in Section 3.1).
We slightly adapted the caption of Figure 1 to make it more clear which data were used.

p. 5, caption Figure 1:
*Time series of a daily mean value of the following parameters: (a) aerosol optical depth (AOD) (in case of a missing daily values, the monthly mean value is shown), (b) integrated water vapour (IWV), (c) ozone (O₃) and (d) surface pressure (p).*

- Section 2.2., lines 136-137, it is not needed to repeat the period included in the analysis.

We removed this sentence about the time period.

- Line 189, Kokhanovsky is good reference, but not the only source where optical properties of clouds are explained. You could add "for example".

We added "for example" as suggested:

p.8, l. 196f:
*"… the optical properties of water and ice clouds is for example given in Kokhanovsky (2004)."*

- Why do you need a section 3.1.1 if there is no section 3.1.2?

We integrated Section 3.1.1 in Section 3.1 and renamed it to "Radiative Transfer Model and its Input Parameters".

- Starting in section 3.1 but then across all the rest of the paper. Be cautious with the use of significant figures in the numbers you give. Many times you give four significant figures, but only one (or two) are really significant. For example, 53.54 % ±20.92 should be 53 % ±21; 5.71 Wm-2 ±11.91 should be 6 ±12, and so on.

Whenever needed, we changed the numbers accordingly throughout the manuscript.

- Lines 233-234. The sentence "As ice habit the default solid column is included" sounds awkward to me.

We changed the sentence to:

p. 9, l. 238f:
*The assumed shape of the ice crystals is a solid column.*

- Section 3.2. I suggest reordering the description of these methods. First, all methods regarding St-As, that is COT and reff; then, methods regarding Ci-Cs, that is COT and SSA.

We rewrote Section 3.2 generally to improve its readability (p. 9, l. 244ff).

- Line 522. In the first moment, it is not clear if DSR refers here to global (or total) irradiance. Later, the reader infers it, as results for direct and diffuse are commented. But I suggest indicating always which irradiance the authors refer to. In addition, I encourage consistency regarding the use of "total" and "global".

We added the term "total" DSR whenever we think it is needed to be more specific.

Throughout the manuscript we changed the terms "global" radiation to "total" radiation.

**- Final comment in section 5.1. It should be discussed the fact that AOD cannot be measured under cloudy conditions. So, strictly speaking, AOD data are not introduced the same way under cloud-free and under cloudy conditions.**

Indeed, we do not use AOD values the same way during cloudy and cloud-free conditions. However, as you also pointed out, it is not possible to measure AOD values under cloudy conditions and therefore under these conditions we do not have another option than to take an averaged AOD value over a longer time period. To be more clear we added:

p. 13, l 343f:
*However, the good agreement in the direct radiation confirms the proper use of the RTM AOD inputs under cloud-free conditions.*

Additionally, we added in the description of the input variable:

p. 8, l. 208ff:
*For cloudy conditions, when AOD can not be measured from the ground or from space, or in cases of missing AOD values during cloud-free conditions, the AOD is a monthly mean value from MODIS data over the whole time period analysed.*

**- Section 6.2. I don't understand one result commented here. If MODISs underestimates COT, these lower COT should result in an overestimation of DSR.**

Thanks for this comment, it was indeed a mistake and it is an overestimation.

p. 20, l. 443f:
*This analysis results in a mean overestimation of $DSR_{MODIS}$ of 88 $Wm^{-2}$ in comparison to the measured DSR during St-As conditions in Payerne.*

**- Conclusions. Your results seem to indicate that COT_modis are not correct, at least not correct under conditions analyzed in the present study. I suggest adding this conclusion.**

We added the following sentence to the conclusions:

p. 21, l. 484f:
*Therefore, we conclude that for a specific location (in this case Payerne) and for high temporal resolution data, $COT_{MODIS}$ are not reliable.*

**Technical matters:**
**- Please check acronyms and their definitions. For example, DSR is defined in the abstract, but not before its first use in the main text. Or, SSA is defined in line 187 but it has been used many times before.**

Whenever needed, we added or removed descriptions of the acronyms.

- Table 3. COT and SSA are dimensionless variables; they don't have Wm-2 units

Thanks for this comment, we removed Wm$^{-2}$ in the caption.